# Development of a New PM Tool Steel for Optimization of Cold Working of Advanced High-Strength Steels

**Abdulbaset Mussa [1],\* , Pavel Krakhmalev [1] , Aydın Şelte [2] and Jens Bergström [1]**

[1] Department of Mechanical and Materials Engineering, Karlstad University, 651 88 Karlstad, Sweden; pavel.krakhmalev@kau.se (P.K.); jens.bergström@kau.se (J.B.)

[2] Uddeholm AB, 683 40 Hagfors, Sweden; aydin.selte@uddeholm.com

\* Correspondence: abdulbaset.mussa@kau.se; Tel.: +46-722-181-367

**Abstract:** In the present study, Uddeholm Vancron SuperClean cold work tool steel was investigated concerning wear resistance and fatigue strength, using laboratory and semi-industrial tests. The Uddeholm Vancron SuperClean was designed with the help of ThermoCalc calculations to contain a high amount of a carbonitride phase, which was suggested to improve tribological performance of this tool steel. In order to investigate the tested steel, galling tests with a slider-on flat-surface tribotester and semi-industrial punching tests were performed on an advanced high-strength steel, CP1180HD. Uddeholm Vanadis 8 SuperClean containing only a carbide phase and Uddeholm Vancron 40 containing a mixture of carbides and carbonitrides were also tested to compare the performance of the tool steels. The microstructure and wear mechanisms were characterized with scanning electron microscopy. It was found that the carbonitrides presented in Uddeholm Vancron SuperClean improved its resistance to material transfer and galling. Semi-industrial punching tests also confirmed that Uddeholm Vancron SuperClean cold work tool steel also possesses enhanced resistance to chipping and fatigue crack nucleation, which confirms the beneficial role of the carbonitride phase in wear resistance of cold work tool steel.

**Keywords:** Vancron SuperClean; cold work tool steels; advanced high-strength steels; sliding wear; galling; punching and chipping

## 1. Introduction

Developments in sheet metals that lead to increasing strength for every generation of sheet metals put high demands on cold work tool steels. Mechanical properties such as high hardness and good toughness are essential for tool steels to withstand the stresses that are generated when forming stronger sheet materials. Tool steels manufactured by the powder metallurgy (PM) manufacturing route have fine and evenly distributed hard phase particles in a relatively tough matrix. Tool steel manufacturers commonly govern steel properties to meet the required demands by combining a hard phase and tough matrix [1–3].

Several studies have prevailed that a higher volume fraction of the hard phase in the tool steel will enhance its resistance to galling and improve its wear characteristics in general [4–7]. However, it was later shown that the volume fraction of the hard phase should not exceed 30% since it reduces the machinability of the tools. It also decreases the tool's toughness and thereby negatively affects its fatigue strength [8]. In recent decades, the introduction of nitrogen as an alloying element into PM tool steels gained attention from tool steel makers. Nitrogen works as a substitute for the carbon and leads to the formation of primary carbonitrides in the steel matrix. Commonly, high nitrogen content in

PM tools is obtained by solid-state nitriding of the steel powder before capsulation. Carbonitrides are generally very fine participates distributed evenly in the tool steel matrix, and, therefore, they contribute to an enhanced toughness of tool steels. Moreover, the carbonitrides have a lower tendency to adhesion during sliding contact and, thus, have a significant impact on the galling and adhesive wear resistance of the tools steels [4,5,9].

Advanced high-strength steels (AHSS) that have an extremely high ultimate tensile strength and limited ductility have introduced challenges in sheet metal-forming applications. Hence, forming AHSS sheets require high press loads to deform or cut them into a final shape. The high press loads result in tough contact conditions between the tool and the work material that often accelerate wear and fatigue damages of the used tools [10]. Wear of tools in cold forming is one of the main concerns since worn tools deteriorate the tolerances, shape, and surface finish of the produced parts. Therefore, worn tools have to be refurbished or replaced by new ones. However, refurbishing worn tools or replacing them leads to production stop and maintenance work, thereby increasing production costs [11]. For that reason, tools with enhanced wear resistance are highly necessary for cold forming.

The wear presented in cold forming results from repeated contact between the tools and work material, and it leads to surface damages of the used tools and deteriorates the quality of the produced part [11–13]. Galling is considered the most common wear mechanism in sheet metal forming. It involves the transfer of work material to the tool surface during the contact between them [14,15]. The galling phenomenon has been widely investigated, and it is subdivided into three stages: The first stage is the initiation of material transfer from the softer surface to the harder, often from the work material to the tool surface. At this stage, the material transfer rate is low, and the sliding contact is between the tool and work material. The second stage is the accumulation of the transferred material. During this stage, the material transfer rate is much higher than in the first stage. In the course of repeated contact, the transferred material is work-hardened and ploughs through the work material, picking up more material and creating microscopic scratches on the workpiece surface. Finally, during the last stage, the transferred material forms macroscopic lumps that result in a high and unstable level of friction between contacting surfaces [4,5,16,17]. Depending on the amount of transferred material, macroscopic scratches can be formed on the produced part, consequently deteriorating its surface and its tolerances. At this point, the surface of the tool is not suitable for further forming process [18].

Material transfer to the tool surface provokes fatigue crack initiation and propagation [19]. Fatigue is a critical and life-limiting phenomenon for punches used in cold forming. D. D. Olsson et al. [20] have investigated the development of pick-up material on the punch surface. It has been reported that the amount of pick-up on the punch surface increases with an increasing number of strokes. Furthermore, the development of pick-ups increased the backstroke force significantly. Therefore, contact stresses at the punch surface locally intensified, and, in many cases, the fatigue strength of the tool material locally exceeded that which resulted in the initiation and propagation of cracks. This means that material transfer to the punch surface will change the surface condition of the tool and thereby influence the quality of the produced part. Moreover, an extensive material transfer will intensify contact stresses at the tool surface during punching. Depending on how high stresses are generated, fatigue crack initiation and growth might take place at the surface. Crack growth due to repeated contact will result in either local failure of the punch (chipping) or total failure. Commonly, the fatigue failure of the tool initiates with the nucleation of microscopic cracks that propagate during further forming process [21]. The fatigue life prediction of tools is difficult because the tool material is subjected to alternating stress conditions as a result of a combined effect of wear and cyclic loads. In order to improve the fatigue life of tools in cold forming, a thorough understanding of material properties such as microstructure and its response to cyclic loading is essential. Therefore, cold forming tools have to be designed carefully concerning their toughness and hardness.

The present study aimed to investigate the influence of the microstructure of three powder metallurgical tool steels, Vancron SuperClean, Vancron 40 and Vanadis 8 SuperClean on their tribological performance in punching applications. To evaluate galling resistance and wear characteristics of these

tool steels, dry and lubricated sliding tests were performed. Finally, semi-industrial punching tests were performed to evaluate the tested tool steels under conditions closest to the real application. To fulfill the aims of the present study, two different testing methods were utilized, a slider-on-flat-surface tribotester, and a semi-industrial punching test method. The evaluation of the tool steels' resistance to galling and fatigue cracking was performed by using advanced high-strength steel sheets.

## 2. Materials and Methods

### 2.1. Materials

Three highly alloyed powder metallurgical (PM) tool steels, Vancron 40, Vancron SuperClean, and Vanadis 8 SuperClean, termed V40, VSC, and V8SC, respectively, were investigated in the present work. The tool steels were hardened and tempered according to the data presented in Table 1.

**Table 1.** The sequence of the heat treatment, modulus of elasticity, toughness, and hardness of the tested tool steels.

| Tool Steel | Austenitizing | | Vacuum Quenching | Tempering | Hardness $(HRC)_{50}$ | Modulus of Elasticity (GPa) | Toughness (J) |
|---|---|---|---|---|---|---|---|
| | T (°C) | Time (min) | Time (s) | T (°C) | | | |
| V40 | 1020 | 30 | 100 | 560 | 831 ± 12 | 236 | 35 |
| VSC | 1080 | 30 | 100 | 540 | 779 ± 14 | 236 | 35.4 |
| V8SC | 1100 | 30 | 100 | 540 | 750 ± 7 | 230 | 33 |

The microstructure of the tested tool steels together with their chemical compositions are presented in Figure 1 and Table 2, respectively. The microstructure of PM tool steels contains a fine and evenly distributed hard phase in a relatively tougher matrix of tempered martensite. V40 contains $M_6C$ carbides, the bright phase, and VCN carbonitrides, the dark phase, as shown in Figure 1a. VSC contained only VCN carbonitrides (Figure 1b), while V8SC had VC carbides as the hard phase (Figure 1c).

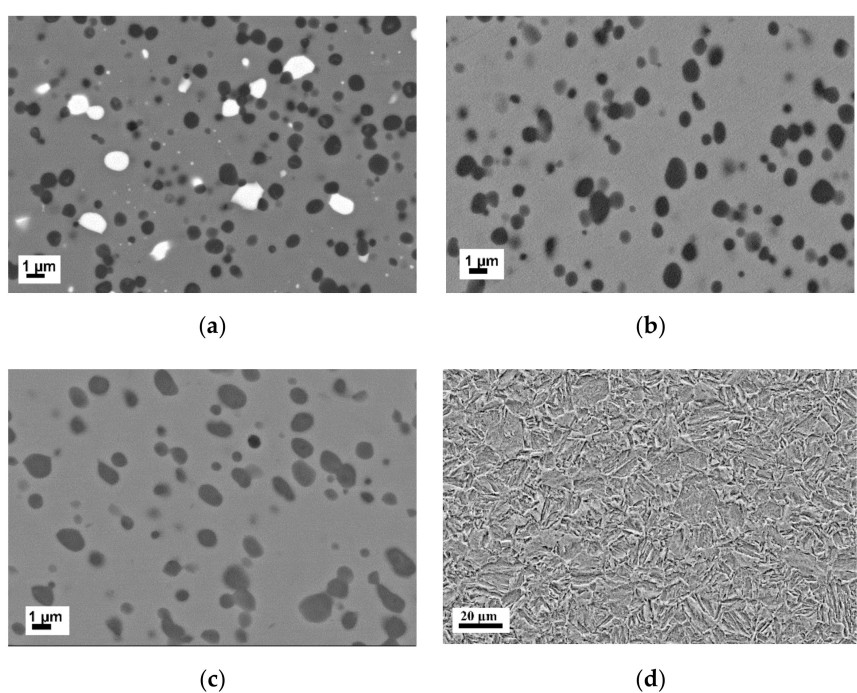

**Figure 1.** The tested tool steels' microstructure and sheet microstructure, (**a**) V40, (**b**) VSC, (**c**) V8SC and (**d**) CP1180HD.

**Table 2.** Nominal chemical composition, wt %, of the tested tool steels.

| Tool Steel | Mn | Mo | C | N | Cr | Si | V | W |
|---|---|---|---|---|---|---|---|---|
| VSC | 0.4 | 1.8 | 1.3 | 1.9 | 4.5 | 0.5 | 10.0 | - |
| V40 | 0.4 | 3.2 | 1.1 | 1.8 | 4.5 | 0.5 | 8.5 | 3.7 |
| V8SC | 0.4 | 3.6 | 2.3 | 0.05 | 4.8 | 0.4 | 8.0 | - |

An AHSS grade, CP1180HD, was utilized to evaluate the galling, wear and fatigue characteristics of the tool steel. The CP1180HD had a complex phase structure containing a fine microstructure of martensite, ferrite and bainite, as shown in Figure 1d. A limited amount of retained austenite of up to 5–15% is also common in its microstructure [22]. The chemical compositions and mechanical properties of the sheet material are presented in Table 3.

**Table 3.** Nominal chemical composition, wt %, and mechanical properties of the sheet material.

| Steel Grade | C | Si | Cr | Mn | Al | Ti | $R_{p\,0.2}$ (MPa) | $R_m$ (MPa) | E (GPa) | $A_{80}$ (%) | Hardness (HV) $_{50}$ |
|---|---|---|---|---|---|---|---|---|---|---|---|
| CP1180HD | 0.23 | 2.00 | 0.1 * | 3.00 | 2.00 | 0.15 ** | 900 | 1180 | 210 | 7 | 354 ± 6 |

\* A total of 0.1 wt % of Cr and Mo combined. \*\* A total of 0.15 wt% of Ti and Nb combined.

### 2.2. Methods

In the present work, blanking punches were investigated, and the focus was on the side surface of the punch in which reciprocal sliding contact between the punch surface and work material takes place. The idea was to investigate the tool steel resistance to material transfer. It was performed in three steps. The first step included dry galling sliding tests. This step was an accelerated laboratory test to quickly assess which tool steel has better resistance to galling. In the second step, sliding wear tests were performed in lubricated conditions and for longer sliding distances. In this step, the surface damages of the tested tool steels were evaluated. In the third step, tool steels were tested in a semi-industrial test rig. This is a scaled-down laboratory test rig that simulates the real industrial application for blanking with regard to contact stresses, punching speed, lubrication, etc. The galling, wear and fatigue characteristics of the tool steels were evaluated by performing wear tests on a slider-on-flat-surface tribotester, and a semi-industrial punching test with the brand name ESSA$^{TM}$. The semi-industrial punching test is a scaled-down laboratory test rig that simulates the real industrial application with regard to contact stresses, stroke rate, lubrication, etc.

2.2.1. Slider-on-Flat-Surface Tribotester (SOFS)

SOFS tribotester has been used successfully for the simulation of wear in sheet metal forming and to rank the galling resistance of tool steels in sliding contact [2,6,7]. It involves sliding of a disc, made of tool steel, under the action of a normal load against a fixed flat surface (Figure 2a). The discs used in the present study had a thickness of 10 mm and major and minor diameters of 50 mm and 20 mm, respectively. The flat surfaces used as the counterface were sheets of CP1180HD with dimensions of 1200 mm × 1000 mm × 2 mm.

Two different sets of sliding tests were performed on the tool steels. The first set of sliding tests was focused on galling, and the second sliding test set was focused on the development of wear mechanisms in sliding contact. In both sets of the sliding tests, the disc was pressed against the sheet surface with a normal load, $F_N$, and forced to slide. The sliding contact mode selected for the present work was reciprocating sliding with a stroke length of 150 mm and a sliding speed of 0.3 m/s. After each stroke, the disc was lifted from the sheet surface and moved 2 mm in the side direction to start with the subsequent stroke, as shown in Figure 2b. The difference between the two test settings was in the duration of the tests and the surface conditions of the sheets. Prior to testing, the discs were ground

and polished as follows: 4 min of wet grinding with a silicon carbide paper of 2400 grit, and 3 min of polishing with a wet cloth and paste containing 3 µm diamonds. A small lathe was used for grinding and polishing of the discs. The final surface finish of the discs was evaluated by a 3D profilometer. A mirror-like surface finish, with a $R_a$ value of around 0.06 µm, was obtained for all tested discs.

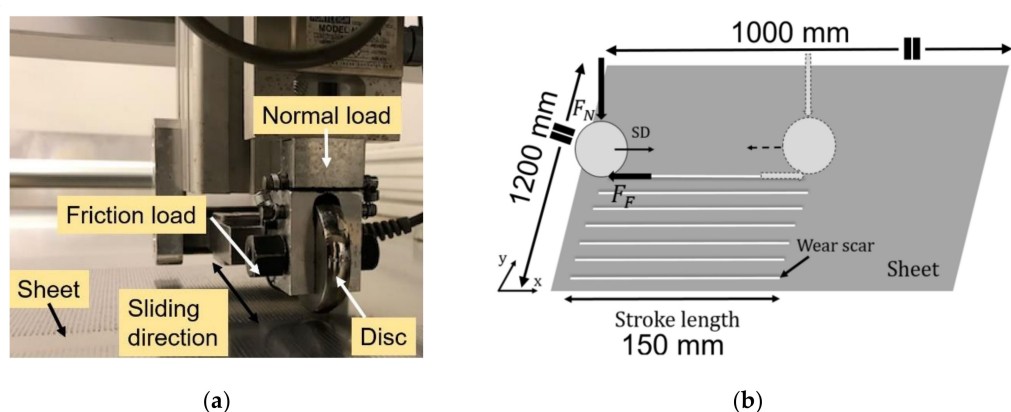

| (**a**) | (**b**) |

**Figure 2.** (**a**) Slider-on-flat-surface (SOFS) tribotester and (**b**) schematic of the performed tests by SOFS. $F_N$ stands for a normal load, $F_F$ stands for friction force and SD indicates sliding direction.

Galling Tests

The galling tests of the tool steels were performed in the dry condition against CP1180HD sheets. Dry tests were accelerated tests performed to quickly assess the resistance of material transfer to the tested tool steels. In this test, the effect of contaminations on galling properties of the tested tool steels was eliminated. To achieve dry contact condition, the sheets were cleaned with a CARMEN PLUS INOX steam vacuum cleaner. Steam together with hot water and a degreasing agent consisting mostly of sodium hydroxide was used to remove the protective oil from the sheet surface. Furthermore, the sheet surface was rinsed with alcohol and wiped with lint-free papers. During each test, the coefficient of friction (COF) between the disc and the sheet was recorded and monitored simultaneously. The COF was studied and was associated with the galling resistance of the tested tool steels. In earlier studies, COF has been used as an indicator of the galling occurrence in sliding contact [23–26]. The galling tests were performed at different normal loads, 50–600 N, and the tests were stopped when the coefficient of friction increased to higher values > 0.5. For each load and each material combination, two test repetitions were carried out.

Wear Tests

Wear tests were performed to investigate the development of wear mechanisms of the tested tool steels after sliding the same distance. The test parameters used were within the range of 50–600 N as the normal load and 450 strokes, 270 m, as the total sliding distance. The wear tests were conducted on the CP 1180 HD sheets in as-delivered surface conditions, i.e., no pre-cleaning of the sheet surface was performed. The surfaces were wiped with lint-free papers to remove the extensive protective oil and also to distribute the remaining oil film at the sheet surface evenly.

Finite Element Method

The static contact pressure between the disc and the sheet was calculated numerically by the finite element method (FEM) using the ABAQUS software (Dassualt Systemes, Velizy-Villacoublay, France). Due to symmetrical geometry of the contact between the disc and the sheet, a model representing only a quarter part of the contact was created in ABAQUS. The C3D10 element type with a 10-node quadratic tetrahedron and controlled hourglass shape was used for the FEM model. The disc was modeled as elastic material and the sheets as elastic–plastic material with a strain hardening behavior according to the stress–strain curve of the CP1180HD.

2.2.2. Semi-Industrial Punching Tests

The wear characteristics of the tools steels were investigated by performing semi-industrial punching tests on CP1180HD sheets. The punching tests were carried out at Uddeholm AB's laboratory, with a 15,000 kg four-pillar eccentric press, ESSA$^{TM}$, as shown in Figure 3a. The eccentric press was equipped with tooling sets for both punching and trimming operations, as displayed in Figure 3b. These two operations can be performed either separately or simultaneously. Note, in the present work, only the punching operation on sheet metal was carried out. The punches used in the present study were cylindrical with a diameter of 9.8 mm and with a polyurethane ejector mounted at the top of the punches, as shown in Figure 4b. The ejector was used to avoid the adherent of the blanks to the top surface of punches. The tested punches were ground to a surface roughness between 0.15–0.2 μm with grinding direction parallel to the cutting edge of the punches. The punches' cutting edge was carefully rounded with a SiC P1200 to a radius of 30–40 μm using a lathe.

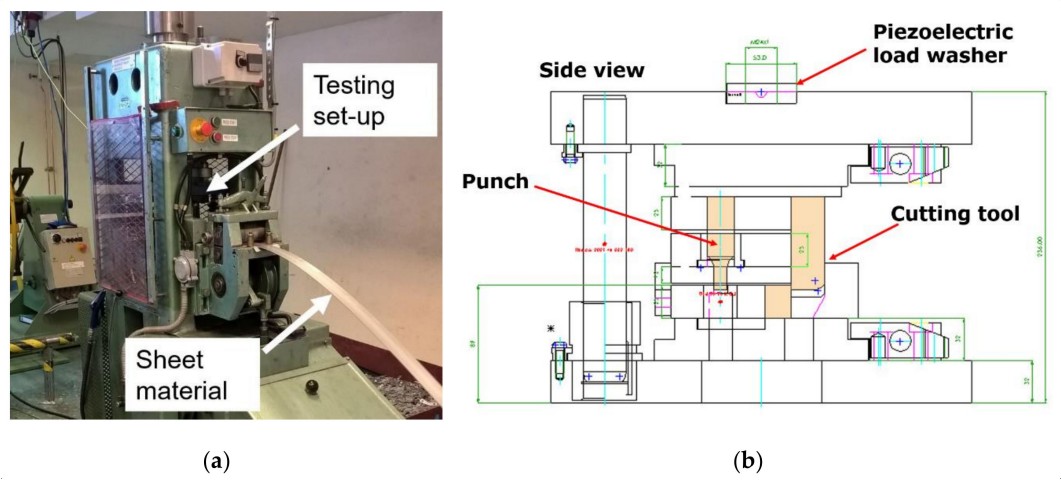

**Figure 3.** Semi-industrial punching test set-up: (**a**) Uddeholm's 15 ton punching test rig and (**b**) a schematic view of the testing set-up.

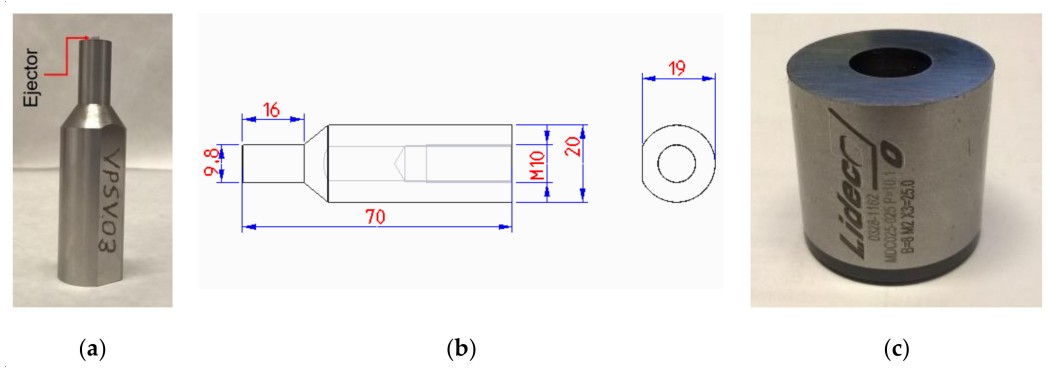

**Figure 4.** Images of the representative test punches and dies: (**a**) punch, (**b**) drawing of the tested punches, and (**c**) die.

Standard dies of MDC025-025, manufactured by LIDECO in Dalstorp in Sweden, with a die clearance of 0.15 mm (Figure 4c) were selected in the present study. Before the punching tests, the roughness of the punches and the radius of the cutting edges were evaluated and verified by a confocal microscope. The punching tests were performed at a stroke rate of 165 strokes/min. The sheet material used for the punching tests had a thickness of 1.5 mm and a width of 50 mm. The punching tests were stopped after 100,000 strokes. This number of strokes was selected based on the previous experimental tests that are sufficient to produce detectable surface damage on the tested punches.

### 2.2.3. Characterization

In order to evaluate the galling resistance and wear and fatigue characteristics of the tested tool steels, a Zeiss-Leo 1530 Gemini field emission scanning electron microscope (Carl Zeiss, Oberkochen, Germany) equipped with the Oxford EDX INCA-sight system (Oxford instruments, High Wycombe, UK) was used to investigate the worn discs and the worn punches. A 3D profilometer, Bruker ContourGT, was employed for the wear evaluation of worn discs and for the measurements of the wear track depth on the sheet surfaces. A confocal microscope, Bruker Contour GT-K (Billerica, MA, USA) and a stereo microscope, Olympus SZ-11 (Olympus, Hamburg, Germany), were used to examine the wear characteristics of the tested punches.

## 3. Results

### 3.1. Microstructure of the Tested Tool Steels

The hard phases presented in the tool steels were different depending on the type, chemical composition, and volume fraction. The V8SC grade contained only the carbide phase. According to the ThermoCalc calculations, V8SC contained 15 vol.% of V-rich carbides. V40 contained two different hard phases, 6 vol.% of $M_6C$ carbides and 14 vol.% of M(C,N) carbonitrides. The VSC was designed to contain 16 vol.% of only M(C,N) carbonitride (Table 4). ThermoCalc data were also used to obtain chemical composition data of carbide and carbonitride phases. The calculations showed that the $M_6C$ carbides in V40 were mainly formed from tungsten, molybdenum, iron, and carbon with a small amount of other alloying elements such as chromium and silicon. Furthermore, the ThermoCalc calculation showed that the VCN carbonitrides presented in V40 and VSC had different chemical compositions. The VCN in VSC had no tungsten in their chemical composition, while VCN in V40 contained 1 wt.% of tungsten. Moreover, the VCN in VSC had lower nitrogen content and a higher amount of iron, molybdenum, and carbon than the VCN in V40. Similarly, the calculations showed that V8SC contained carbides as the hard phase particles in the matrix that were formed mainly from vanadium, molybdenum, chromium, iron and carbon (Table 4). It should be mentioned that the hard phases presented in the tested tool steels have a face-centered cubic lattice structure [1].

**Table 4.** Chemical composition wt% of the hard phase and its volume fraction presented in the tested tool steels.

| Hard Phase | V | N | C | Mo | Fe | Cr | Mn | Si | W | Volume Fraction % |
|---|---|---|---|---|---|---|---|---|---|---|
| VCN in VSC | 73.8 | 13.7 | 5.9 | 2.8 | 2.2 | 1.6 | $5.3 \times 10^{-3}$ | $4 \times 10^{-8}$ | 0 | 16 |
| VCN in V40 | 73.9 | 15.6 | 4.4 | 1.4 | 1.9 | 1.6 | $4.9 \times 10^{-3}$ | $9 \times 10^{-9}$ | 1 | 14 |
| M6C in V40 | 0.3 | 0 | 2.3 | 24.8 | 33.6 | 4.2 | 0 | 0.76 | 34 | 6 |
| VC in V8SC | 60.0 | 0.4 | 14.7 | 16.2 | 2.9 | 5.7 | $2 \times 10^{-2}$ | $7 \times 10^{-7}$ | 0 | 15 |

To understand the significance of the microstructure on galling and wear characteristics of the tested tool steels, the average diameter of the hard phases presented in each tool steel and the average distance between the particles were analyzed. SEM was used to measure the diameter of the hard phase particles. For each tool steel grade, two samples were used, and on each sample, a diameter of 30 particles was measured. Later, the average diameter of the hard phase particles was calculated. The measurements were taken in different locations on the sample to obtain an accurate average diameter for the particles. The average distances between the hard phase particles were also calculated by measuring the distance between 50 adjacent particles for each sample. It was found that the largest particles were presented in V40, and they were $M_6C$ with an average diameter of 1.4 μm. The smallest particles were VCN in V40 with an average diameter of 0.81 μm. The second largest particles were VC presented in V8SC with an average diameter of 1.21 μm, and the carbonitride hard phase particles presented in VSC had an average diameter of 0.9 μm, as shown in Figure 5a.

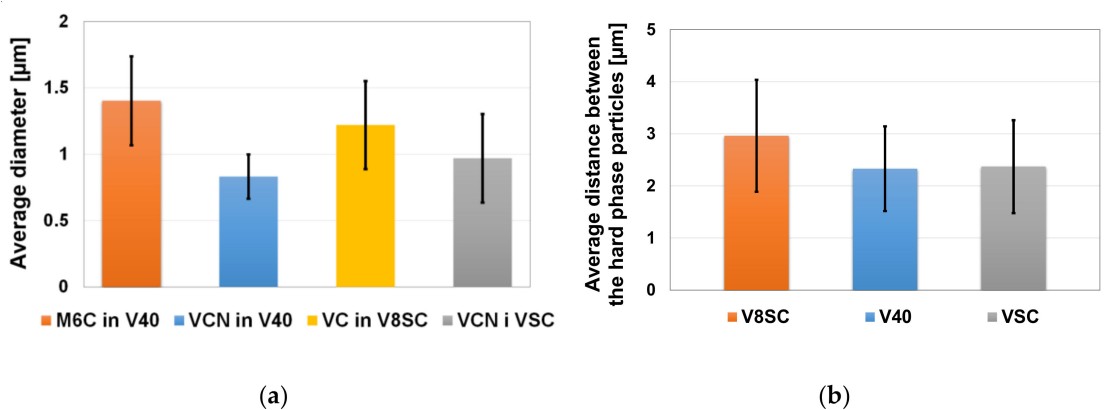

(**a**)                                                    (**b**)

**Figure 5.** Measurements of the hard phase presented in the tool steel matrices: (**a**) the average diameter of the particles and (**b**) the average distance between the hard phase particles.

The shortest average distance between the hard phase particles was found in V40 matrix with a mean distance of 2.33 μm. The second shortest distance was presented in the VSC matrix with an average distance of 2.37 μm, whereas the longest distance was found in the V8SC matrix with 2.96 μm (Figure 5b).

*3.2. Galling*

The galling tendency of the tested tool steels was investigated by using the COF as an indicator of the galling occurrence. The COF for each test was monitored as a function of the sliding distance. It was observed that the COF for all tests started with a low and a stable value <0.2. During further sliding it increased to a higher value >0.5 with an instable behavior, as shown in Figure 6. The increase in COF to high and unstable values was considered as the occurrence of galling. Changes in friction were associated with extensive material transfer, i.e., galling phenomena. The sliding distance required for the tool steel to reach galling initiation was considered as the critical sliding distance, and it was measured from the COF curves. Furthermore, the critical sliding distance for each test was plotted against the test maximum contact pressure.

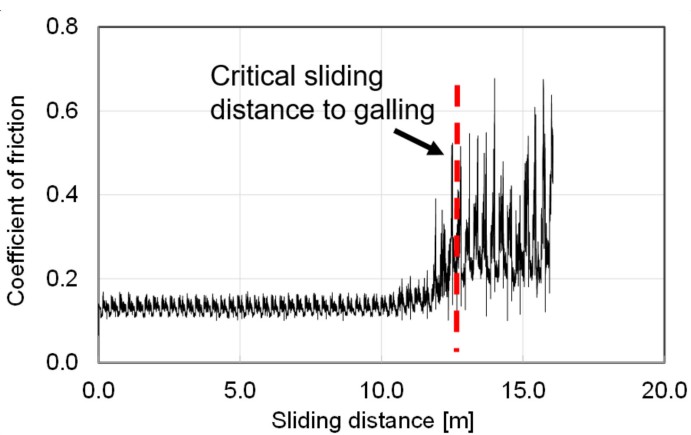

**Figure 6.** The coefficient of friction (COF) as a function of sliding distance for V40 tested against CP1180HD at 60 N (860 MPa).

The summary of the galling tests is presented in Figure 7. The galling tests showed that for all the three tool steels, galling resistance decreased with the increased contact pressure. The trend was noticeable despite the tool steel grade. However, for the tests performed at the same contact pressure, it was found that the critical sliding distance to galling was dependent on the grade of the tool steel. From the galling diagram, it was observed that V40 and VSC showed better galling resistance than

V8SC. This behavior was more pronounced at lower contact pressures (Figure 7). With increased contact pressure, it was observed that V8SC still had the shortest critical sliding distance to galling. Nevertheless, at higher contact pressures, the critical sliding distance for all the three tested tool steels decreased, and, therefore, the trend of better galling resistance of V40 and VSC was less noticeable.

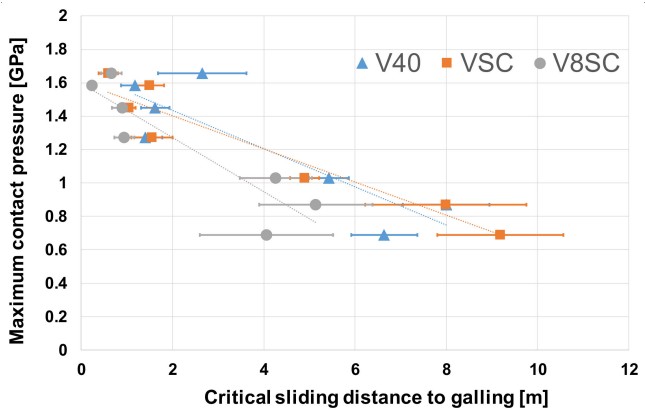

**Figure 7.** The summary of the galling tests for the tested tool steels.

### 3.3. Wear Characterization

The wear characteristics of the tested tool steels were investigated by studying the worn surfaces exposed to a total sliding distance of 450 strokes, 270 m, at different contact pressures in a range of 0.69–1.77 GPa. It was observed that worn areas with an almost elliptical shape were formed on the discs' surfaces as the result of sliding contact between the discs and the sheet material, and the worn area was covered by adhered material from the sheets.

Furthermore, the worn area for each test was investigated with SEM. At low magnification, similar wear characteristics for all the three tool steels were observed. The dominant wear mechanism was adhesive wear and transfer of sheet material to the tool steel surface, as shown in Figure 8. It was also observed that the adhered material at the worn surface was not evenly transferred. To some extent, the adhered material was accumulated into a thicker layer at the center region of the contact area at the disc surface. However, in general, there were locations at the worn surface where the tool surface was completely covered by the adhered material, but there were also locations where only a thin layer of adhered material could be detected. Nevertheless, these two distinct behaviors were stochastically presented at the worn surface.

SEM investigations carried out at higher magnification revealed wear mechanisms at the micro-scale. Depending on the contact pressure of the tests, the development of surface damage was observed. For the discs tested at contact pressures below 1 GPa, the worn surface of the tested tool steel was sheared, and adhesive wear was the dominating wear mechanism. The partial detachment of hard phase particles from the steel matrix was observed. The minor detachment of carbides from the tool steel matrix was observed for V40 and V8SC tool steels. As the testing contact pressure increased, the surface material was sheared more severely, and adhesive wear was observed for all the tested steels. Additionally, some particle detachment was revealed, and the severity of this wear mechanism depended on the tool steel grade. For VSC and at high contact pressures, the dominant wear mechanism was adhesive wear with a minor detachment of carbonitride particles from the matrix, as shown in Figure 9a. For V8SC, the partial and fully detachment of the hard phase particles was easily detected, as displayed in Figure 9b. The V40 steel was worn the most, and full detachment of the VCN presented in V40 was found at the worn surface. Moreover, it was observed that the $M_6C$ particles at the worn surface were fractured into smaller pieces. Besides detachment and fracture of the hard phase in V40, micro-fatigue and cracks were observed at the worn surface. These cracks were propagated perpendicular to the sliding direction, as shown in Figure 9c.

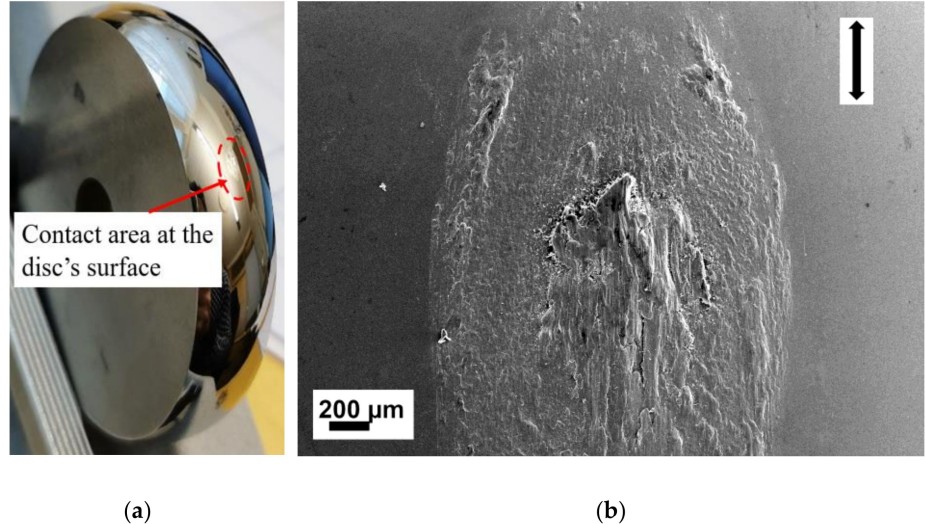

**Figure 8.** The typical worn surface of the discs after the occurrence of galling: (**a**) image of the disc and (**b**) SEM image of the contact area.

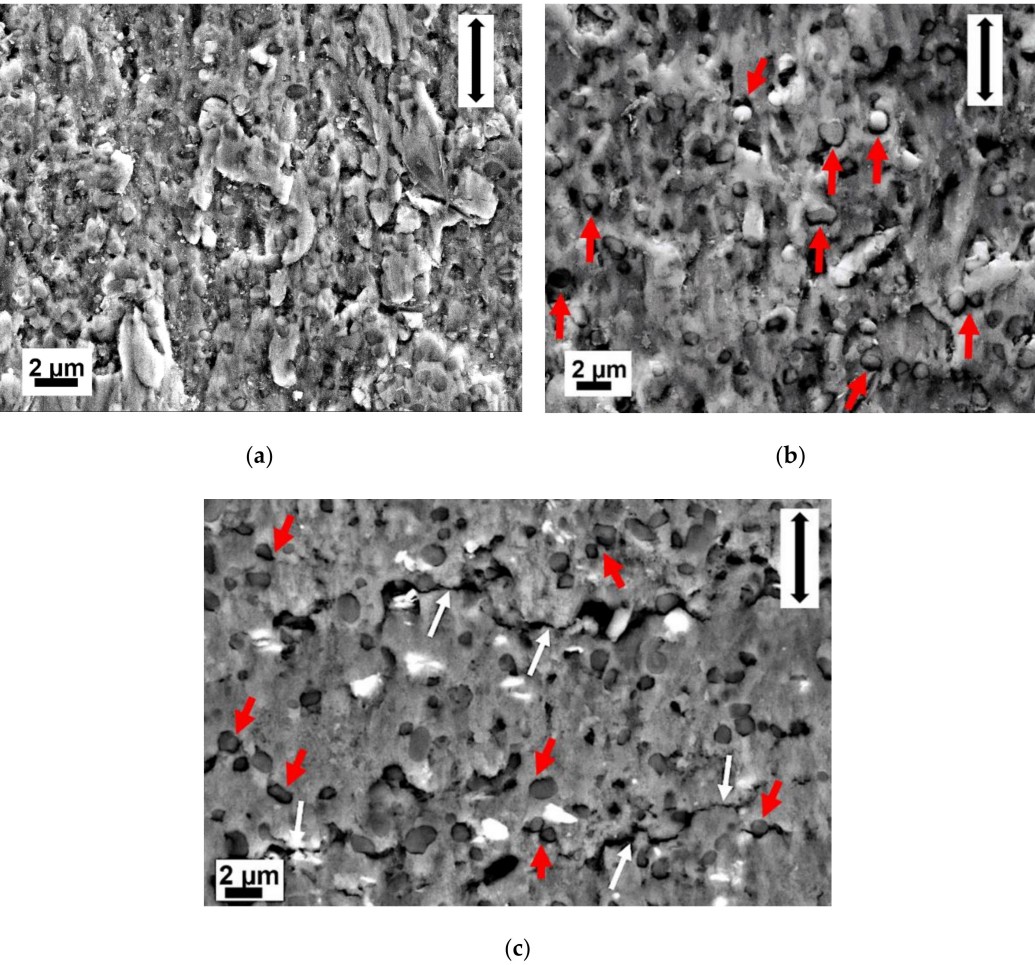

**Figure 9.** The worn surface of the discs tested against CP1180HD at 500 N (1.65 MPa) and 450 strokes: (**a**) VSC, (**b**) V8SC and (**c**) V40. White arrows indicate the fatigue cracks, and red arrows indicate the hard phase detachment.

To reveal the severity of surface deformation and behavior of hard phases beneath the surface, cross-sectioned discs were investigated. It has been revealed that the hard phase particles presented in V40 deformed severely during sliding contact under high contact pressures. It was observed that M6C particles at and beneath the worn surface were fractured into smaller particles. In contrast, the VCN particles were found to be elongated along the sliding direction, as shown in Figure 10a,b. Contrarily, the hard phases presented in VSC and V8SC did not change in shape even after sliding at high contact pressures. The VCN in VSC and VC in V8SC did not show any evidence of fracture or plastic deformation like those observed in the V40. Fatigue cracks formed during sliding contact were also observed when the cross-sections of VSC and V8SC were investigated. However, the cracks were found to be formed at the interface between the tool surface and the adhered material. The propagation of the cracks was found to be in the transferred material, as shown in Figure 10c,d.

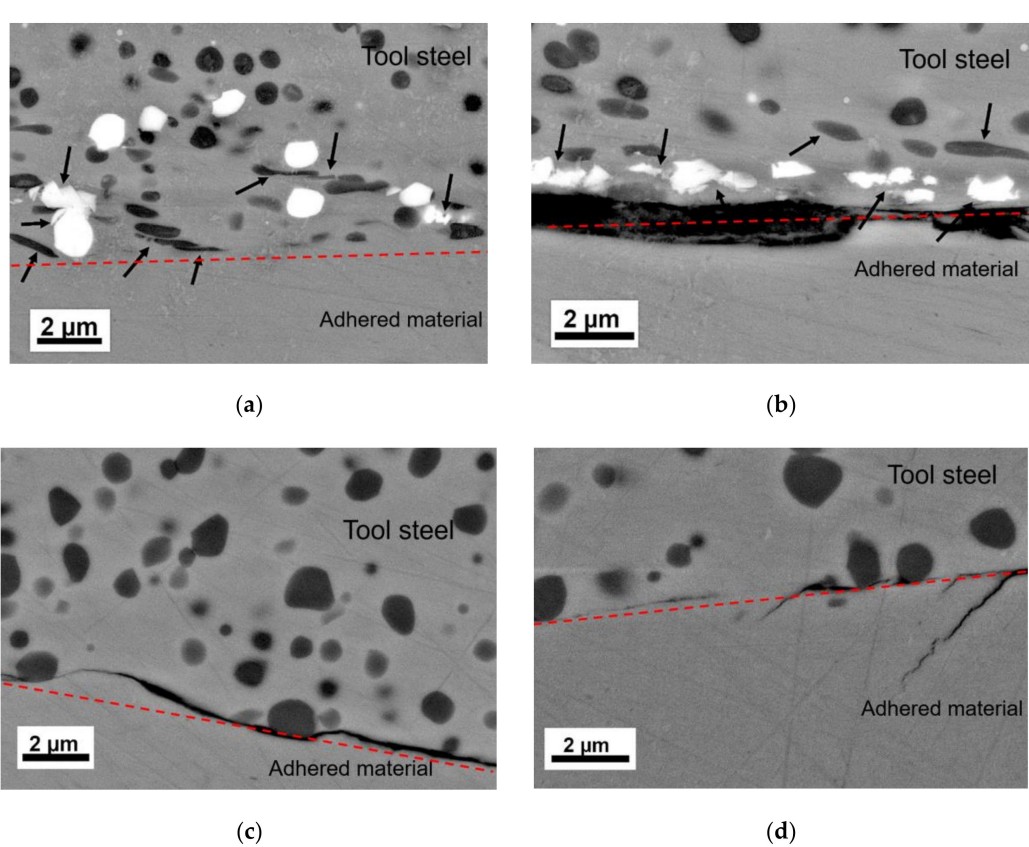

**Figure 10.** SEM images of the cross-section of the worn tools tested at 500 N (1.65 MPa) and 450 strokes against CP1180HD sheet: (**a**) V40, (**b**) V40, (**c**) VSC and (**d**) V8SC. Arrows indicate the deformed and fractured hard phase particles.

### 3.4. Punching Tests

As a result of the lower performance of the hard phase presented in V40 during sliding contact, only VSC and V8SC were tested in the semi-industrial punching tests. After running 100,000 strokes of punching CP1180HD for each punch, the tested punches were first examined using the stereomicroscope. It was observed that side surfaces of the punches picked up sheet material as a result of sliding contact between the punch surface and sheet material. Additionally, the amount of pick-up material was found to be dependent on the type of tool steel. For the VSC punches only a thin layer of sheet material was transferred to the side surface of the punches Figure 11a, whereas a higher amount of the material pick-up was observed for the V8SC punches Figure 11b. It was also noticed that the pick-up depth on the side surface of the punches was larger for the V8SC punches. The pick-up depths for VSC and V8SC punches were around 1.8 and 3.4 mm, respectively. The pick-up moved successively backward with

respect to the punching direction while increasing the number of punching strokes. Therefore, the more pick-up gave a larger depth at the punch surface. In addition to the material transfer, other damage mechanisms such as fatigue cracks and chipping were observed. The fatigue cracks were associated with chipped tool steel material from the punch cutting edge. However, the chipping phenomenon was only observed on tested punches made of V8SC tool steel. Chipping results in geometrical changes of the contact between the sheet and the punch, and it limits the punch performance during the forming process. At the cutting edges where macro chipping was observed, the extensive material pick-up was not found compared to the location where no chipping was observed, as demonstrated in Figure 11b.

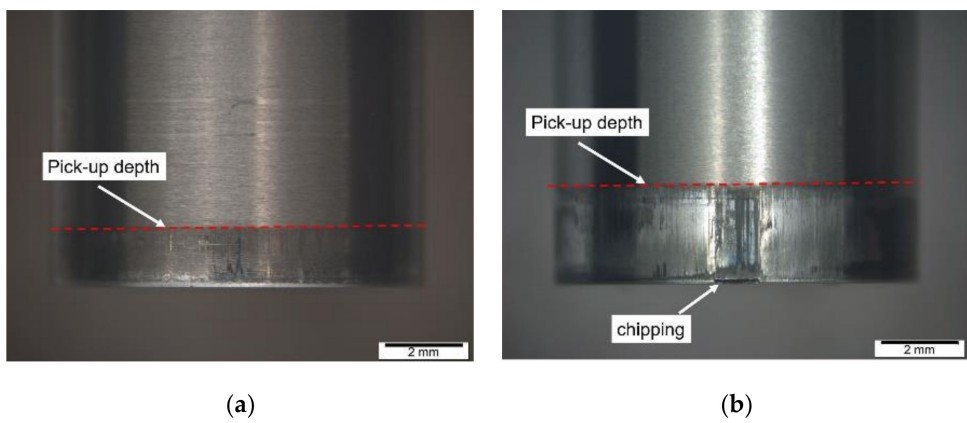

|(**a**)|(**b**)|

**Figure 11.** Worn punches after 100,000 strokes against CP1180HD: (**a**) VSC punch and (**b**) V8SC punch.

VSC punches made the 100,000 strokes without any detectable chipping, and, thus, it was further investigated at high magnification by SEM. It was found that almost all grinding marks were removed from the punch surface after the contact with the sheet material (Figure 12a). Picked up material was observed at the side surface. Moreover, it was observed that tool material was removed from the cutting edge and the side surface of the punches due to adhesive wear (Figure 12b). Micro-scratches due to abrasive wear were also observed at the worn surface of the punch. The abrasive scratches were parallel to the punching direction and occurred mainly in the matrix material around the hard phase particles (Figure 12b). The cutting edge for the VSC punches was less worn and without any chipping throughout the entire test duration of 100,000 strokes. VSC punches were further evaluated by the 3D profilometer. It was found that a certain rounding of the cutting edge had occurred after 100,000 strokes. A comparison of the initial cutting edge and worn cutting edge shape of punches made by the VSC punches is illustrated in Figure 13a,b. The edge rounding of worn punches is associated with plastic deformation of the edge material, and micro-abrasive wear occurred during punching.

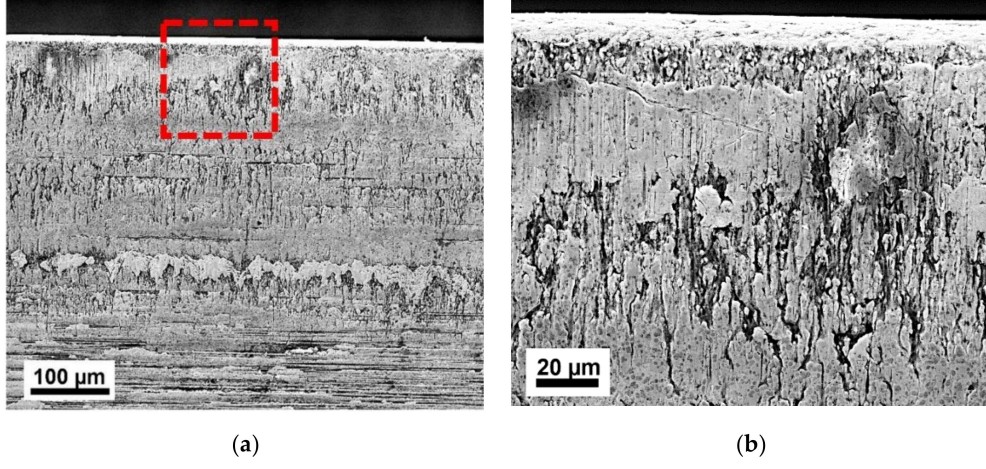

|(**a**)|(**b**)|

**Figure 12.** (**a**) Worn surface of the VSC punch and (**b**) a higher magnification of the cutting edge.

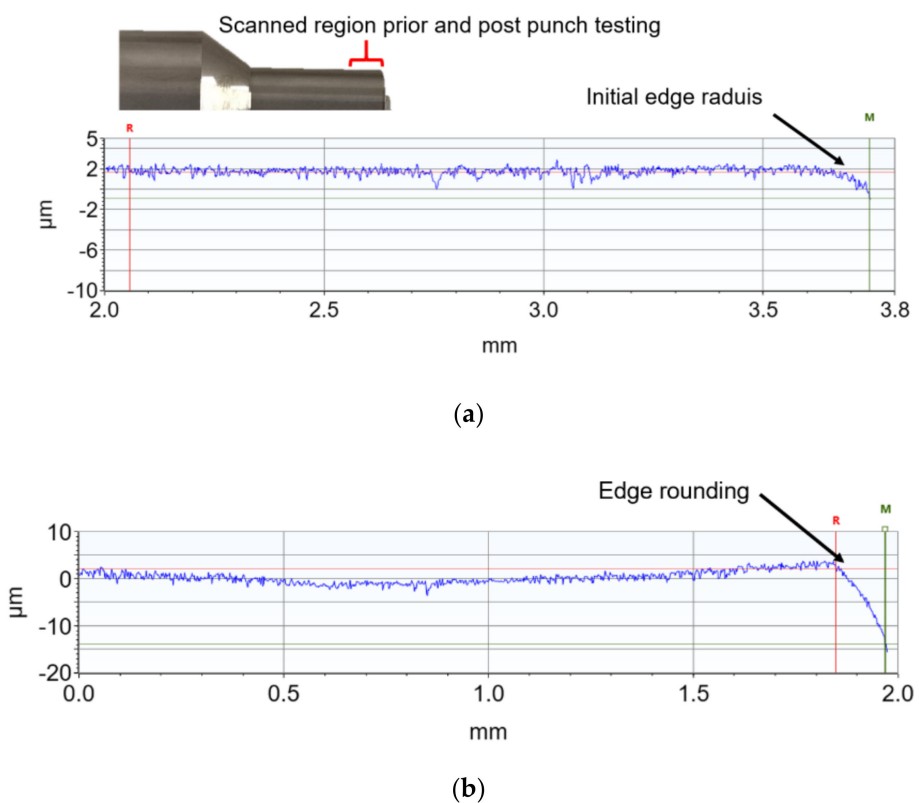

(**a**)

(**b**)

**Figure 13.** 3D profilometer analysis of the VSC punch: (**a**) unworn prior to punching testing and (**b**) worn punch after 100,000 strokes.

## 4. Discussion

Wear and fatigue are life-limiting mechanisms for punches used in cold forming applications. In many cases, these two damage mechanisms might take place simultaneously, resulting in a synergetic effect that leads to a catastrophic failure of the punches. Therefore, it is important to take the resistance of a material to wear and its ability to withstand cyclic loads into consideration when designing cold forming tools. In the present study, it was found to be of interest to investigate the galling, wear, and fatigue characteristics of a newly developed cold work tool steel, VSC. The Uddeholm Vancron SuperClean cold work tool steel was designed with help of ThermoCalc calculations to contain a high amount of a carbonitride phase, which was suggested to improve tribological performance and wear resistance of this tool steel.

Two other cold work tool steels, V40 and V8SC, were also investigated in the present work. The results regarding galling resistance of the tested tool steels showed that the critical sliding distance to galling decreased with increased contact pressure, as shown in Figure 7. The influence of contact pressures on galling occurrence in cold forming has been studied by other researchers [2,7,12–15]. It has been reported that the galling resistance of tool steels can be improved by decreasing the contact pressure between the tool and the work material surfaces [7]. Nevertheless, when forming AHSS sheets, high contact pressures are not avoidable due to the high strength of these steel grades [2,14]. Therefore, material transfer and eventually galling cannot be prevented in the cold forming of AHSS. However, adequate knowledge about the selection of proper tool steel for a particular application is the key to reducing galling related problems in the application. The presented results showed that the tool steels with carbonitrides VCN had better resistance to galling compared to the tool steel containing VC carbides. This behavior was pronounced for the sliding tests performed at contact pressures up to 1.5 GPa, as shown in Figure 7. At contact pressures higher than 1.5 GPa, the critical sliding distance to galling was short for all the three tested tool steels. Therefore, the trend of better galling resistance for the tool steels with VCN was less pronounced.

Galling is the result of material transfer from one surface to another when the surfaces are exposed to the relative motion under certain pressures. The galling phenomenon is sensitive to many factors such as microstructure and properties of the contacting surfaces, roughness, temperature, the chemical composition in the interface between surfaces, etc. [6,7,15,27,28]. In the present work, it was found to be of high importance to eliminate factors that affect the transfer of material between contacting surfaces such as lubrication and roughness. Therefore, the galling tests were performed in the dry contact condition. Moreover, the discs' surfaces were polished to a mirror-like surface, around Ra = 0.06 μm, in order to eliminate any extensive initial ploughing of sheet material caused by tool surface roughness. In an earlier study, it was reported that the tool surface has different morphologies at the nano-scale depending on the type of hard phases presented in the tool matrix. It has also been revealed that even a nano-scale roughness has a significant impact on galling. Hence, the nano-scaled protrusions plough through the sheet surface, displacing or picking up material at the sheet surface [5]. P Karlsson et al. [29] measured the height of the hard phase protrusions using an atomic force microscope. They found that VCN particles protrude up to a 25 nm in height, while $M_6C$ and VC protrude up to about 5 nm. They suggested that at low contact pressures, the real contact is between these protrusions and the sheet material, and the protruding hard phase particles are the main elements that pick up sheet material. During sliding contact, the protrusions plough through the sheet material, and a pick-up material around the protrusions occurs. According to this statement, the more and the higher protrusions, the higher amount of transferred material. However, in the present study, it was found that the tool steels that contain VCN have less tendency to pick up sheet material. This trend was more significant at lower contact pressures. As the contact pressure increased, the tendency of sheet material to stick to the tool surface became similar for all the tested tool steels. It is believed that at low contact pressures, the real contact is between the tool surface protrusions and sheet material, but this does not mean that every protrusion will contribute to the pick-up of the sheet material. The protrusion might plough through the sheet surface and displace material, leaving behind micro-scratches. It seems that the chemical composition of the hard phase that affects the affinity of sheet material to stick to it is more critical than its height. I. Hekkilä et al [4] studied the frictional behavior of different hard phases commonly presented in tool steels. They found that the VC particles led to higher frictional forces than the VCN particles. This is in good agreement with the experimental results observed in the present study. From the galling diagram, it is seen that V8SC has a shorter critical sliding distance to galling than the V40 and VSC. The V8SC steel contains VC particles that probably contributed to higher frictional forces between the tool and the sheet material, and thereby a higher rate of material transfer to the tool surface was obtained.

Other factors that have contributed to a better galling resistance found for V40 and VSC in the present work are the particle size and distance between them. It is well-known from earlier studies that smaller hard phase particles in a tool steel and shorter distance between them improve galling resistance [2]. The measurements of the particle size showed that the carbonitride VCN phase had the smallest size compared to the other carbides, $M_6C$ and MC. $M_6C$ presented in V40 had the largest size, 1.4 μm. However, the volume fraction of $M_6C$ in V40 was only 6%, that is, the dominant hard phase in V40 was not $M_6C$, but it was VCN that had the smallest size of 0.81 μm. The number of hard phase particles and their size determines the distance between the particles. V40, due to its highest volume fraction of the hard phase of 20%, had the shortest distance between its particles (2.33 μm) followed by VSC, where the average distance between particles was 2.37 μm. The small VCN particles presented in V40 and VSC and the short space between them can be associated directly with their good resistance to galling. VSC compared to V40 had a lower volume fraction of the hard phase (16%) and still had almost the same average distance between the hard phase particles as in V40. This is due to the small size of VCN particles in VSC. V8SC that showed a lower galling resistance had a longer average distance between its particles, with a particle mean size of 1.2 μm.

With the occurrence of galling, the material accumulation takes place more rapidly, and complex tribological situations arise between contacting surfaces. Therefore, in the present work, it was found

of great interest to run wear tests to a predetermined sliding distance without stopping the tests, even when galling has occurred. It enables the comparison of wear mechanisms of the tool steels tested under similar contact conditions. The results from the wear tests revealed that wear mechanisms were dependent on the type of material and on the contact pressures. At low contact pressures, it was found that the worn surface was plastically deformed as a result of shear stresses generated during contact. In addition, the worn surface was covered by transferred material from the sheets. Additionally, it is believed that sheet material continuously transfers to the tool surface during sliding contact. Depending on the strength of the interface between the transferred material and the tool surface, the transferred material might stick and accumulate during contact, resulting in a high amount of material pick-ups, or, in the case of a weak interface, the transferred material can detach from the tool surface. It might also lead to a situation where tool material together with transferred material is removed from the tool surface due to adhesive wear. The worn surface of the tested tools consisted in regions where the worn surface was totally covered by adhered material and regions with less amounts of adhered material, and, in such regions, other wear mechanisms like abrasive scratches at the tool worn surface were observed, as shown in Figure 8.

Moreover, hard phase particles presented in the tested tool steels were partially detached from the steel matrix. As the disc slides against the sheet material and due to the presence of high frictional forces, high shear stresses are generated at the tool steel surface. Subsequently, the subsurface of the tool is plastically deformed, as shown in Figure 10. The tool steel matrix and its hard phase do not have the same strain limits; therefore, hard phase particles might detach from the matrix material. The detachment of hard phase particles could also be dependent on the adhesion forces between particles and the steel matrix.

At higher contact pressures, it was observed that the tested surface was severely worn, and detachment of hard phase particles for V40 and V8SC was more pronounced. For V40, for instance, full detachment of the hard phase particles, VCN, and fracture of $M_6C$ particles were revealed by SEM analysis at high magnification, as shown in Figure 9c. In addition, fatigue cracks at the worn surface were also observed for V40. Further examination of the cross-section near the worn surface revealed that VCN particles in V40 were elongated along the sliding direction, and $M_6C$ were broken into smaller pieces. The deformation of the hard phase particles confirms the occurrence of high shear stresses during sliding contact. Breakage of hard phase particles could be the reason for the initiation of fatigue cracks during sliding contact. The formation of the fatigue cracks in the present tests could also be related to low-cycle fatigue. During the reciprocal sliding contact, and due to the presence of high shear stresses, accumulation of strains in the subsurface material takes place. According to the literature [30], the softer material subjected to cyclic loads at high stresses might accommodate higher strains and, thereafter, be more resistant to fatigue crack initiation [30]. This hypothesis is in good agreement with the observed cracking behavior of the tool steels in the present study since no fatigue cracks were found for the VSC and V8SC that have a lower hardness compared to V40. The capacity of these two tool steels to accumulate a higher amount of strains during cyclic plastic deformation without cracking could be one reason why fatigue cracks were not found for them. Moreover, it was also noticed that even at high contact pressures of up to 1.7 GPa, the fracture or deformation of the hard phase particles did not take place for these two tool steels.

The elongation of V40 hard phase particles, VCN was observed due to plastic deformation. Contrarily, the VCN particles in VSC showed high stability under high contact pressures. The reason could be due to the chemical composition of these particles since the VCN in VSC does not have the same chemical composition as VCN in V40, as shown in Table 4. The VCN in VSC has a higher content of carbon, molybdenum and iron compared to the VCN in V40. It has also a lower amount of nitrogen. The selected chemical composition of the VCN in VSC probably provided a better performance of the VSC tool steel when it was exposed to high pressure sliding contact. Therefore, it showed great strength under high contact pressures. The VC in V8SC also showed a high strength under high-pressure sliding contact. However, detachment of these particles from the tool matrix

became more pronounced at higher contact pressures, as shown in Figure 9b. The detachment of the VC in V8SC can be associated with the interaction between steel matrix and the hard phase particles. During reciprocal sliding contact, the surface material is repeatedly sheared as the disk slid forward and backward. Due to plastic deformation, strains are accumulated at the surface material of the disc. The hard phase particles in the softer matrix act as discontinuities for the plastic flow of the tool material. This will result in strain localization in the tempered martensitic matrix around the VC particles, leading to the detachment of these particles during further sliding. I. Picas, et al. [31] studied the microstructural effects on fatigue crack nucleation in cold work tool steels. In their work, it is reported that fatigue occurred due to the presence of voids that served as crack initiation sites. The voids were created during cyclic loading as strains were localized around the carbides, resulting in the detachment of carbides from the matrix.

Results from the semi-industrial punching tests regarding galling resistance were in a good agreement with the results from SOFS tests. In both cases, the VSC tool steel with VCN particles showed a better galling resistance compared to V8SC tool steel containing carbides. The superior galling resistance of VSC can be associated with its hard phase friction characteristics against sheet material in addition to the hard phase size and its homogenous distribution in the tool matrix. It was found that after 100,000 strokes, the punches made of VSC had only a thin layer of transferred material at a depth of 1.8 mm, as shown in Figure 13a. In contrast, punches made of V8SC had more adhered material. Furthermore, the adhered material on V8SC punches had a depth of up to 3.4 mm, as shown in Figure 13b. The development of the material pick-up on the side surface of the punches was investigated earlier [20]. It was found that the pick-up on the punches developed successively backward from the head of the punches with an increased number of strokes. In the present study, it is believed that material from the sheet was continuously transferred to the punch surface, and further punching resulted in the movement of the pick-up material in the opposite direction of the punching direction. The more pick-up material, the higher the depth of the transferred material. Adhered material on the cutting edges and side surface of the punches alternates the local stress conditions at the punch surface at the contact with the sheet material. During repeated punching, and due to the adhered material, the stress level at the surface might raise and cause the initiation of microcracks. The interaction between the hard phase particles and the martensitic matric can contribute to the initiation of fatigue cracks. Here, in the sliding tests, it was noticed that the detachment of the hard phase particles from the matrix occurred in V8SC steel. This can act as crack initiation sites during the cyclic loading. Further punching will result in the propagation of the initiated microcracks, and, eventually, pieces of punch material will chip out. The stress level is dependent on the amount of adhered material and the adhesion of the interface between the adhered material and the punch surface. As was observed in the present work, the chipping phenomenon occurred only on the V8SC punches. A combination of alternation in stress level at the punch surface due to transfer material and the interaction between the hard phase and martensitic matrix seems to be the main reason for the chipping of V8SC punches, as it is well-known that chipping is highly correlated with mechanical properties of the punch material such as hardness and toughness. A careful balance between these two properties might decrease the risks for chipping during punching [21]. However, VSC and V8SC have slightly similar hardness and toughness. Therefore, it is believed that the amount of adhered material to the punch surface and detachment of carbides were critical for the chipping of the V8SC punches.

## 5. Conclusions

The galling resistance, wear mechanisms, and fatigue characteristics of three different PM tool steels were investigated by performing dry and lubricated SOFS sliding tests and semi-industrial punching tests.

- The dry SOFS tests, and especially at low contact pressures, showed that tool steels with VCN particles showed better resistance to galling than the tool steel contained VC. With increased contact pressure, the resistance to galling for all three tools became rather similar. Smaller hard

phase particles with a shorter distance between them seem to improve the resistance of the tool to galling at low contact pressures.

- In the wear SOFS tests, the hard phases presented in V40 had less stability under high pressure sliding contact. The hard phase particles in V40 were deformed and fractured. In contrast, hard phases presented in the other tested tool steels showed no signs of deformation. The carbide fracture observed for V40 served as fatigue cracks initiation site, and fatigue crack growth due to repeated contact took place. No fatigue cracks were observed for V8SC or VSC.
- Concerning galling resistance, SOFS test results were in good agreement with the test outputs from semi-industrial punching tests. In both tests, VSC showed a better galling resistance than the other tested tool steels. In semi-industrial punching tests, the galling resistance of VSC was superior when compared to V8SC. In the punching tests, chipping and fatigue cracks observed for V8SC punches were the result of extensive material pick-up and the localized strain accumulations around the carbides.

**Author Contributions:** A.M.: conceptualization; methodology, investigation; validation; visualization, writing the original draft; A.Ş.: conceptualization; investigation; resources, validation, writing, review and editing; P.K.: supervision, project administration, funding acquisition, review and editing; J.B.: supervision, project administration, funding acquisition. All authors have read and agreed to the published version of the manuscript.

**Funding:** This research was funded by the Swedish knowledge foundation, project no 20150090.

**Acknowledgments:** The funding of the present research work and the material supply by Uddeholm AB is gratefully acknowledged.

**Conflicts of Interest:** The authors declare no conflict of interest.

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
