# Peer review of "Development of a New PM Tool Steel for Optimization of Cold Working of Advanced High-Strength Steels"

_metals, doi:10.3390/met10101326_

Round 1

Reviewer 1 Report

Increasing use of high-strength steels for lightweight applications require enhanced tool steels for their economically reasonable processing. The paper deals with the effect of alloys in cold working steels and the formation of carbides on adhesive wear mechanisms during blanking of advanced high-strength steel. Adhesive wear mechanisms must be assessed and investigated scientifically in order to enable upcoming kinds of enhanced tool steels for blanking processes. An overall interesting and promising work.

In order to improve the readability and scientific character, some changes and additional explanations are suggested. In general, the paper is too long and unstructured, especially in the introduction and discussion sections.

General comments:

  • The title should not contain a trademark name and AHSS should be fully spelled.
  • English level regarding spelling and grammar should be improved.
  • It is unclear, which process is referred to. The authors mention that the investigated tool steels are used for forming processes. However, only a blanking process was the subject of the tests. The translation of the findings to different forming processes is not justified and doubted because the tribosystem strongly differs in other forming processes like deep drawing or extrusion processes.
  • Statements are often repeated which strongly reduces readability and increases the length of the paper.
  • The methodology section could better support why the methods have been used. Explanation for methodology is presented in the results section instead. The methodology section should answer the questions which cause-and-effect interactions are investigated and how results are obtained and evaluated.
  • The latest cited references are from 2014. Please add the latest references in the field of adhesion wear from the past five years.

Formal comments:

  • The reference list is not consistently formatted.
  • General rules of typography must be revised: the use of spaces, colons, etc.
  • Font size in figures should be partially increased as small fonts are hard to read.
  • Fully spell the header of chapter 2.2.1 (no abbreviation)

Figures and tables:

  • Table 3: The line break in the material is confusing
  • Table 4: what means “V. of hard phase”? The dot in line 2, col 1 makes no sense.
  • Figure 2: symbols must be explained in the figure
  • Figure 5: font size is too small. As a deviation is given, how are the values averaged and how many samples were used?
  • Figure 6: Please add a grid to the diagram. X-axis is labelled with “critical sliding distance to galling”. The critical sliding distance to galling was not defined and the label of the x-axis seems not correct.
  • Figure 7: How were the deviations obtained? How many repetitions of the test were performed?
  • Figure 8: Which view of the disc is presented in this figure? An overview of the disc would be helpful.
  • Figure 9: Show sliding direction.
  • Figure 10: Indicate tool steel and adhered material.
  • Figure 13: Axes must be labelled. Where was the profile measured on the punch? A small graphical explanation would be helpful to understand the plots.

Content:

  • Introduction is too long. E.g. the history of PM-steels has no relevance for the paper.
  • Line 53: It is mentioned that carbonitrides enhance toughness and influence galling and adhesive wear resistance. How are these effects explained and supported?
  • Line 56: Which kind of strength is referred to? tensile strength?
  • Line 65: Surface damage on the tool or on the produced part? If on the tool: to what extent?
  • Line 73: The effect of wear on forming dies on the part quality is referred to. However, blanking punches are investigated in the paper. The effect of punch wear on the part is only relevant in piercing, not in cutting out. If cutting out is the reference process then the given information is not relevant.
  • A scientific problem statement is missing in the introduction section. Which physical cause-and-effect interactions have been investigated?
  • Line 106: By which criteria have these tool steels been selected for investigation?
  • Line 128: What is meant by semi-industrial punching test?
  • Line 131: It is stated that SOFS tribotester is used as sheet metal forming analogy. How can the results be transferred to the investigated blanking process as the tribosystem is not directly comparable? Which is the analogy between SOFS and blanking?
  • Line 134: Please point out the described dimensions in Figure 2.
  • Line 147: Evaluation by naked eye is not verifiable and therefore not scientific.
  • Line 151ff: Why have the galling tests been repeated in dry state (galling) and with anti-corrosive (wear)? Why could wear not also be evaluated in the dry test? Furthermore, it is unclear why anti-corrosive is used instead of forming lubricant.
  • Line 170: The FEM was introduced as a method but never evaluated and used for the explanation of gained results. It remains also unclear how the contact pressure is needed for explanations later in the work.
  • Line 181: Why is there a trimming operation in the tool and does it have an effect on the blanking operation? Please explain.
  • Line 183: A PU-ejector is mentioned. Where is this ejector located in Figure 3b? Does the ejector have an influence on tool wear or the load collective in the punch? Is the cutting tool in Figure 3b the mentioned trimming tool?
  • Line 232: Two terms are used to describe the size of carbides: equivalent diameter and mean size. How has the equivalent diameter been measured and averaged, can a standard deviation or a variation been given, and how many samples have been used for mean size? How was average distance measured?
  • Line 246: The increase of the COF is explained by occurrence of galling without the timepoint of occurrence could be measured directly. How is the observed effect explained?
  • Line 259: What is the definition of the critical sliding distance?
  • Line 265: The unit of a distance is meter.
  • Line 279: Use the impersonal form.
  • Line 288: Detachment of hard phase particles is described. How was this observed? How can detachments be observed in figure 9 by the reader?
  • Line 321: Sheet material thickness in the blanking test is mentioned to be 1.5 mm. How can wear marks be seen up to 1.8 resp. 3.4 mm from the cutting edge? This sounds like a systematical influence from differing tool conditioning in both cases. Chipping has only been observed at the punch with the larger wear zone (longer penetration depth of the punch). It is questionable if chipping can only be explained by the punch material condition if experimental conditions were not kept constant.
  • Line 350: Again, cold forming is referenced. However, the tribosystem differs strongly e.g. in cold extrusion, deep drawing and shearing processes. This must be distinguished.
  • Line 362: Give the most relevant reference.
  • Line 442: How was plastic deformation observed?
  • Line 475: Does the disc slide in two ways? This must be clarified and justified in the methodology section.
  • The conclusion section is too long and contains many repetitions. It is suggested to discuss cause-and-effect interactions instead of the observations step-by-step.
  • Line 531: Please give an explanation why short distance between hard phase particles improves galling resistance.

Author Response

Dear reviewer,

      Thanks a lot for your valuable comments regarding our manuscript entitled “Development of a new PM tool steel for optimization of cold working of Advanced high strength steels”. We have considered your comments and carefully made all corrections, we hope the revised version of our manuscript meets your approval. Please see the attachments.

best regards

Abdulbaset Mussa

Reviewer 2 Report

The article is very interesting. The paper focuses wear of cold work tool steel. The experimental and microscopic wear mechanism approaches show that authors are advanced in the both approaches. The results obtained are of industrial novelty and practical value. At the same time, some additions and corrections are needed:

(1) First, there is a lack of references recently published, which could undermine the reliable analysis of the scientific value of the current study. Therefore, some of the articles recently published needs to be referred to in the revised manuscript to provide clear information to the readers.

(2) Application of the steel examined, CP1180, is needed.

(3) More detail description of SOFS is needed

(4) Meaning of MDC025-025 in line 190

(5) Correct the unit of contact pressure in line 266 MPa -> GPa

Author Response

Dear reviewer,

Thanks a lot for your valuable comments. The manuscript has been revised according to the provided comments. Please see the attachment it is the revised version of the manuscript.

(1) First, there is a lack of references recently published, which could undermine the reliable analysis of the scientific value of the current study. Therefore, some of the articles recently published needs to be referred to in the revised manuscript to provide clear information to the readers.

Some relevant articles published between 2018 and 2020 have been referred to in the revised manuscript.

(2) Application of the steel examined, CP1180, is needed.

Application of CP1180 has been added in the material and method part.

(3) More detail description of SOFS is needed

To keep the method part reasonably large and to avoid abundant information, SOFS has been described briefly, and references that describe SOFS in details has been given.

(4) Meaning of MDC025-025 in line 190

it is a standardized die made of M2 steel grade manufactured by Lideco.

(5) Correct the unit of contact pressure in line 266 MPa -> GPa

The unit is corrected in the revised version.

Best regards

Abdulbaset Mussa

Round 2

Reviewer 1 Report

Comments have been adequately considered. The quality of the paper was significantly improved.